# ON PRE-TRAINING OF MULTIMODAL LANGUAGE MODELS CUSTOMIZED FOR CHART UNDERSTANDING

## ABSTRACT

Recent studies customizing Multimodal Large Language Models (MLLMs) for domain-specific tasks have yielded promising results, especially in the field of scientific chart comprehension. These studies generally utilize visual instruction tuning with specialized datasets to enhance question and answer (QA) accuracy within the chart domain. However, they often neglect the fundamental discrepancy between natural image-caption pre-training data and digital chart image-QA data, particularly in the models' capacity to extract underlying numeric values from charts. This paper tackles this oversight by exploring the training processes necessary to improve MLLMs' comprehension of charts. We present three key findings: (1) Incorporating raw data values in alignment pre-training markedly improves comprehension of chart data. (2) Replacing images with their textual representation randomly, during end-to-end fine-tuning, transfers the language reasoning to chart interpretation skills. (3) Requiring the model to first extract the underlying chart data and then answer the question in the fine-tuning can further improve the accuracy. Consequently, we introduce CHOPINLLM, an MLLM tailored for in-depth chart comprehension. CHOPINLLM effectively interprets various types of charts, including unannotated ones, while maintaining robust reasoning abilities. Furthermore, we establish a new benchmark to evaluate MLLMs' understanding of different chart types across various comprehension levels. Experimental results show that CHOPINLLM exhibits strong performance in understanding both annotated and unannotated charts across a wide range of types.

## 1 INTRODUCTION

In today's data-driven world, visualizations like bar and pie charts are crucial for deciphering complex datasets. However, the increasing diversity and complexity of these charts highlights the need for advanced tools to enhance human capabilities in data analysis. Artificial Intelligence (AI), particularly Multimodal Large Language Models (MLLMs), is increasingly used to automate the understanding of scientific charts, promising more efficient and accurate analysis. Robust benchmarks are also essential, setting standards and metrics that drive the development and evaluation of these AI tools.

Prior studies have introduced end-to-end neural models aimed at enhancing chart comprehension (Lee et al., 2023; Liu et al., 2022b; Zhou et al., 2023), such as masked table prediction (Zhou et al., 2023), chart question answering (Masry et al., 2023), and chart de-rendering (Liu et al., 2022b). These models specialize in handling one task each within the domain of chart analysis. Furthermore, advancements in Multimodal Large Language Models (MLLMs), exemplified by LLaVA (Liu et al., 2024b; 2023) and miniGPT (Zhu et al., 2023), have showcased their versatility in vision-language tasks. These generalist models undergo a two-stage training process: initially learning visual-language alignment through image-caption pairs, followed by end-to-end fine-tuning using image-QA pairs. This training not only enables LLMs to interpret visual data but also retains their extensive pre-trained knowledge, which supports their reasoning abilities and leads to strong performance across diverse visual language understanding tasks.

Recent advancements have further ignited interest in tailoring MLLMs to specialized domains such as scientific chart understanding. Han et al. (2023); Liu et al. (2024a) have explored collecting instruction-tuned chart data and low-rank adaptation (Hu et al., 2021) to enhance MLLMs' proficiency with unique chart characteristics. However, research on the fundamental-training regimes – namely,

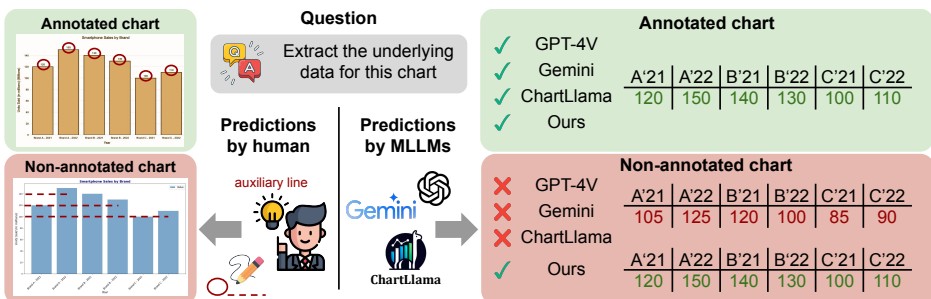

Figure 1: The underlying data values can be inferred regardless of whether the chart is annotated. However, existing MLLMs rely on annotations and struggle with unannotated charts. In contrast, our model bridges this fundamental discrepancy between natural image-caption pre-training data and digital chart image-QA data, enabling it to extract values regardless of whether the chart is annotated.

pre-training to align across modalities and comprehensive end-to-end fine-tuning – for chart-specific understanding remains scarce. As shown in Fig. 1, existing MLLMs often struggle to extract the underlying data from charts when numerical values are not annotated. We hypothesize that this issue stems from a gap in vision-language alignment between natural image-caption pairs and digital chart-data pairs. Without targeted pre-training for chart-data alignment, models may resort to relying on a "shortcut" of recognizing numeric annotations through OCR during fine-tuning with QA pairs, rather than truly understanding the visual subtleties of diverse charts.

This paper addresses the above issues by concentrating on the essential training methodologies for MLLMs, including cross-modal feature alignment pre-training and comprehensive end-to-end fine-tuning. Our research is guided by the question, *"How does fundamental MLLM training influence the enhancement of general MLLMs with chart-specific domain understanding?"* Our findings indicate that: (1) Raw data extraction are pivotal in alignment pre-training to bolster chart data comprehension; (2) Substituting some chart images with purely textual data during end-to-end fine-tuning not only preserves LLM's text-only reasoning ability but also augments chart interpretation capabilities; (3) Augmenting QAs with data extraction tasks in the fine-tuning phase allows model to achieve the data prompting during testing, where it first extract data and then answer the QAs, further improving the its reasoning skills. Furthermore, existing chart benchmarks are limited in chart and question types. This motivates us to introduce a comprehensive chart benchmark, comprising 20 chart types and three QA levels, to better measure MLLM performance and advance future research in this field.

Our key contributions are summarized as follows:

- We introduce CHOPINLLM,[1] a Multimodal Large Language Model tailored for comprehensive chart understanding. This model excels at interpreting various chart types including unannotated ones, underpinned by our detailed analysis and training guidance that emphasizes the importance of foundational training for chart-specific tasks.
- We propose a novel data generation pipeline using text-only Large Language Models to efficiently produce large-scale pairwise data. This approach significantly reduces the costs and complexity of data generation for MLLM training.
- We establish a robust benchmark comprising a diverse array of chart types and question-answering levels, designed to rigorously evaluate MLLMs' fundamental understanding of the scientific chart domain.

## 2 RELATED WORKS

**Large Language Models (LLMs).** LLMs have seen remarkable advancements in recent years, primarily driven by transformers (Vaswani et al., 2017) that allowed significant scaling in model size and training data (Chowdhery et al., 2023; Brown et al., 2020; Du et al., 2022; Dai et al., 2019; Fedus et al., 2022; Hoffmann et al., 2022; Rae et al., 2021; Smith et al., 2022). These models excel in generalized reasoning and exhibit robust chain-of-thought reasoning (Wei et al., 2022; Wang

---

[1] **Ch**art **O**riented **P**retraining **In**tegration in **L**arge **L**anguage **M**odels

et al., 2022; Zhang et al., 2023c) across a variety of tasks, largely attributed to extensive pre-training (Devlin et al., 2018; Zhao et al., 2023; Beltagy et al., 2019) and fine-tuning strategies (OpenAI, 2023a; Ouyang et al., 2022; Chung et al., 2024). The availability of powerful LLMs with specialized capabilities – ranging from general assistance (Gemini Team, 2023; OpenAI, 2023b; Anthropic, 2023; Touvron et al., 2023) to coding (Roziere et al., 2023; GitHub, 2023; Jiang et al., 2023) – has fueled diverse applications such as data augmentation (Ding et al., 2024), data generation (Yu et al., 2024; Patel et al., 2024), and providing training guidance (Yuan et al., 2024; Kwon et al., 2023). These developments have markedly accelerated research and practical applications in the field.

**Multimodal Large Language Models (MLLMs).** Building on the success of LLMs, recent research has expanded their application to multimodal tasks, including image (Liu et al., 2024b; Zhang et al., 2023b; Lu et al., 2024; McKinzie et al., 2024), video (Zhang et al., 2023a; Chen et al., 2023a), audio or speech (Fathullah et al., 2024; Das et al., 2024; Borsos et al., 2023), mixed-modal (Team, 2024), various tool and API usages (Patil et al., 2023; 2024; MeetkAI, 2024), and robotics (Zeng et al., 2023; Brohan et al., 2023). In extending LLMs to image modalities, early studies combined LLMs with external vision models to convert visual information into text, enhancing image comprehension (Liu et al., 2022a; Yang et al., 2022). Others have integrated visual encoders directly within LLM frameworks, developing end-to-end systems that transform images into textual tokens (Zhu et al., 2023; Liu et al., 2024b; 2023; Aiello et al., 2023; Driess et al., 2023; Chen et al., 2023b). While maintaining capabilities like reasoning and chain-of-thought processing across various tasks, these models often fall short in domain-specific tasks like chart analysis (Masry et al., 2022; Methani et al., 2020). This prompts further research into specialized data collection and fine-tuning for distinct domains.

**Chart Understanding.** Current approaches to chart understanding fall into two main categories: models specifically designed for chart-related tasks (Lee et al., 2023; Zhou et al., 2023; Masry et al., 2023; Liu et al., 2022b; Masry & Hoque, 2021), and those that utilize pre-trained LLMs and MLLMs (Han et al., 2023; Xia et al., 2024; Masry et al., 2024a; Liu et al., 2024a; Masry et al., 2024b; Meng et al., 2024). The first group involves models trained exclusively on chart-specific data, often limited by the scope of the training datasets thus cannot be applied to diverse chart scenarios. The second group, which involves adapting existing LLMs and MLLMs through fine-tuning (Liu et al., 2024b) or integration with external models (Liu et al., 2022a), shows promising versatility across various questions and scenarios. Yet, there is a scarcity of research on MLLMs' pre-training, crucial for deep chart understanding and adaptability to multiple chart types in practical settings. Additionally, chart understanding models are evaluated against benchmarks focused on tasks like data extraction (Masry et al., 2022; Kantharaj et al., 2022a; Shi et al., 2024), summarization (Kantharaj et al., 2022b), and basic mathematical reasoning (Methani et al., 2020), which predominantly feature basic chart types (*e.g.*, bar, line, pie charts) and lack nuanced differentiation in QA levels to thoroughly assess models' understanding capabilities. Addressing these gaps, our work not only explores effective pre-training strategies for MLLMs on chart data but also introduces a new benchmark with a variety of chart types, differentiated QA levels (*e.g.*, literal, inferential, reasoning), and raw data to evaluate MLLMs' comprehension abilities. Concurrently, CharXiv (Wang et al., 2024) is proposed for evaluating general understanding of real-world scientific charts, including complex compositions with multiple subplots. In contrast, our benchmark focuses on single-plot chart images, evaluating the raw data understanding and mathematical reasoning of an MLLM.

## 3  GENERATING DATA FOR CHART UNDERSTANDING

To build a chart understanding MLLM and study its fundamental training process, a comprehensive dataset containing chart images paired with captions and raw data is essential for pre-training, alongside different types of question-answer pairs for end-to-end fine-tuning. However, no existing dataset provides the necessary variety of chart types, topics, and styles. To bridge this gap, we introduce a novel data generation pipeline for large-scale chart data generation (Sec. 3.1) and QAs generation (Sec. 3.2). With the data at hand, we then explore various training strategies in the later sections, including feature alignment pre-training and end-to-end fine-tuning for LLMs. Figure 2 presents an overview of our framework.

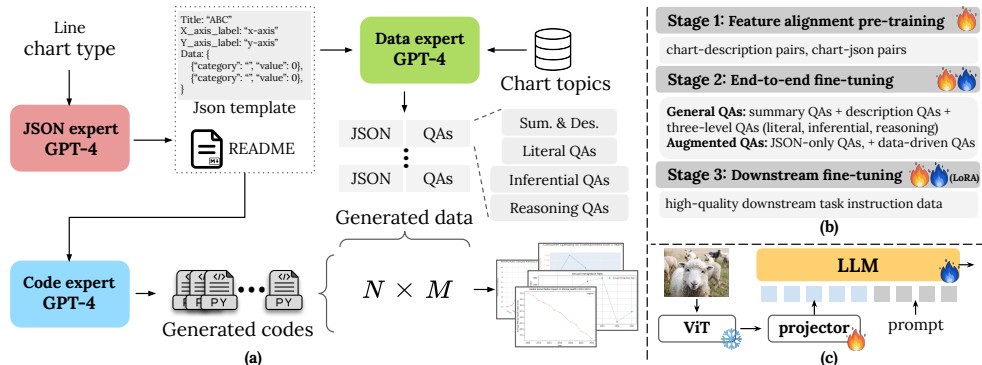

Figure 2: **Overview of (a) the proposed data generation pipeline and (b) training strategies of** CHOPINLLM. Generating code and data points conforming to a shared JSON template enables quadratic scaling of the data size (w.r.t. to #GPT calls). The 3-stage training equips our model to grasp the underlying data, thereby achieving a fundamental understanding of charts. ($N$ and $M$ denote the number of generated scripts and data, respectively.)

## 3.1 EFFICIENT DATA GENERATION WITH QUADRATIC SCALING

Our data generation leverages the promising text content generation and coding abilities of large language models, *e.g.*, GPT-4, to generate chart images and data. Specifically, LLMs allow us to synthesize raw data for chart images, and then the generated Python script turns the raw data into a chart image. In this way, we can produce image data without accessing costly multimodal LLMs like GPT-4V. Unlike previous and concurrent works (*e.g.*, Han et al. (2023); Xia et al. (2024)) that prompt LLMs to iteratively generate CSV data, QAs, and Python script for each chart image – a process that is costly to massively scale – our pipeline features parallel code and data generation through shared templates and READMEs for consistent definitions and formats across the same chart types. Most importantly, since all code script and data share the same structure, our generated data can be universally applied to any generated code and vice versa, significantly enhancing scalability without exhaustedly prompting LLMs. We detail the pipeline further below.

**Shared Template and README.** As shown in Fig. 2 (a), given a chart type (*e.g.*, line) sampled from a predefined chart type database, the JSON expert GPT-4 first generates a JSON template for the given chart type, along with a README file. In detail, the JSON template contains general information for the chart image, including the title, x-axis, y-axis information, and raw data. The README contains the definition of the chart type and the meanings of the keys and values to enhance understanding of the JSON template. Please refer to Appendix G for some examples. We note that the JSON template, together with the README, ensures the consistency of data generation so that further data and code generation can follow the explicit format and definition guidance of the template data. Note that we choose JSON as our primary data representation format, in contrast to previous works Han et al. (2023); Masry et al. (2022); Methani et al. (2020); Xia et al. (2024), which used CSV. The JSON format allows us to incorporate not only numerical data but also additional chart information, such as titles and the scales of x and y axes, which is beneficial for pair-wise pre-training tasks. Moreover, JSON data is structured, and when paired with a README file, it minimizes ambiguity in data descriptions, which is particularly valuable for complex chart types. For instance, in candlestick charts, we can clearly define a data point as a dictionary containing "open", "close", "high", and "low" values, rather than a list where the meaning of each number might be unclear.

**Orthogonal Data and Code Generation.** With the template files at hand, we can generate data and code independently. For the data generation branch, to ensure the generated data covers diverse topics, we jointly input the produced template files (*i.e.*, JSON template and README) and a topic sampled from a pre-defined topic set (*e.g.*, energy production and market share) into a data expert GPT-4 module. For the complete topic list, please refer to Appendix H. We require the data expert GPT-4 to follow the definitions in the template files and generate $M$ JSON data along with different kinds of questions and answers (*e.g.*, summary QA) based on the raw data. As for code generation, another code expert GPT-4 is utilized to produce $N$ Python code based on the given chart type, data template, and Python library. Note that to prevent generating simple code repeatedly for the given

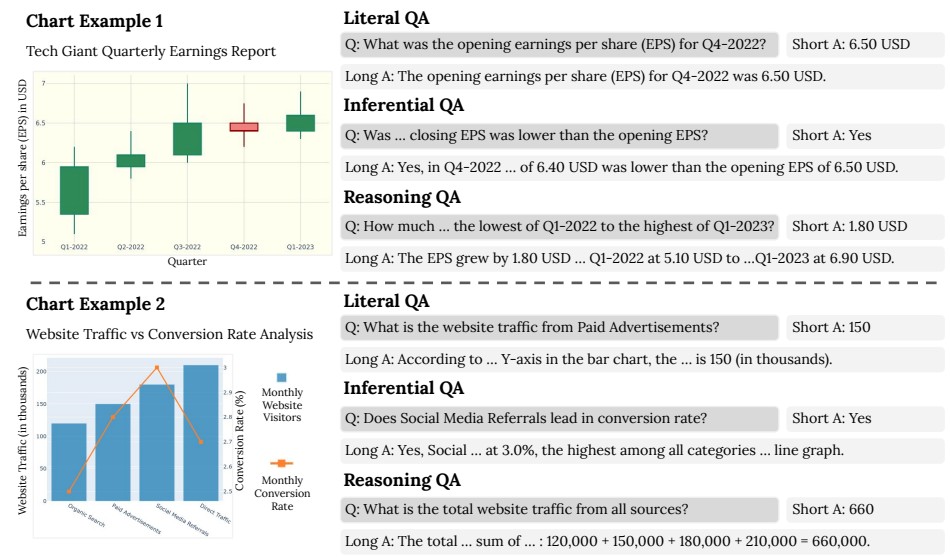

Figure 3: **Examples of generated three-level QAs with long and short answers**, accessing the understanding of charts from various perspectives. Best viewed in color.

chart type, we explicitly ask the code expert GPT-4 to introduce visual variations in aspects such as color, legend, grid, font, and mark texture, *etc*. More details can be found in the Appendix.

## 3.2 DIVERSE QA SYNTHESIS

Based on the parallel data generation pipeline, we are able to collect massive amount of chart image and JSON raw data pairs for the feature alignment pre-training. Now, we details how we generate different types of QAs for end-to-end fine-tuning. Specifically, having each JSON data as input, we use text-only LLM to generate question-answer (QA) pairs. To cover various question-anwser for chart data, we include general QAs, containing not only description and summary QA but also three different level of QAs: literal QAs, inferential QAs, and reasoning QAs (as illustrated in Fig. 3). Furthermore, to enhance the training of chart understanding, we introduce two additional augmented QAs (for training only): text-only QAs and data-driven QAs. We detail each QA type as follows:

- **Description QAs:** Generate objective descriptions based on the chart data.
- **Summary QAs:** Summarize the chart, highlighting key findings.
- **Literal QAs:** Extract specific values directly from the data.
- **Inferential QAs:** Infer global insights, such as identifying extreme values.
- **Reasoning QAs:** Perform calculations to derive answers from chart data.
- **JSON-only QAs:** Replace images with JSON raw data to augmented previous QAs.
- **Data-driven QAs:** Prompt the model to extract JSON raw data before answering the question.

These QAs encompass a range of questions for chart images, covering abilities from basic data understanding and global concept comprehension to advanced reasoning, allowing us to further assess the abilities of MLLMs. Note that, for each QA pair, we use GPT-4 to generate both long and short answers. The long answer, generated first, includes a step-by-step explanation to derive the answer, while the short answer, generated later, contains only the final answer derived from the long explanation. Short answers contain only numerical values or Yes/No response for convenient evaluation purpose. For more examples of generated chart and QAs, please refer to Appendix K.

**Composition for Quadratically Scaled Data.** As shown in Fig. 3 (a), we consider 20 different chart types. For each chart type, we collect 400 different Python codes ($N = 400$) and 1000 different JSON data files ($M = 1000$) covering various topics. Note that we exclude bad data based on predicted file structure's correctness, Python code execution errors, and OCR tools. Please refer to the supplementary materials for detailed information. After filtering, we have approximately 5 million images, with all the chart types listed in Fig. A6. For each chart image, we collect the

Table 1: Comparative analysis with existing benchmarks for chart understanding evaluations. * denotes unbounded chart types. Chart variation refers to whether the dataset contains chart images with different styles but sharing the same raw data.

| Benchmark | # Image | # Chart type | Avg. # QAs per image | Multi-level QAs per image | Raw data per image | Chart Variation |
|---|---|---|---|---|---|---|
| PlotQA Methani et al. (2020) | 33.7k | 3 | 1 | ✗ | ✗ | ✗ |
| ChartQA Masry et al. (2022) | 1.5k | 3 | 1 | ✗ | ✓ | ✗ |
| Chart-to-text Kantharaj et al. (2022b) | 6.6k | 6 | 1 | ✗ | ✗ | ✗ |
| MMC Liu et al. (2024a) | 2k | 6 | 1 | ✗ | ✓ | ✗ |
| Chartbench Xu et al. (2023) | 2.1k | 9 | 9 | ✓ | ✗ | ✗ |
| ChartX Xia et al. (2024) | 6k | 18 | 1 | ✗ | ✓ | ✗ |
| CharXiv Wang et al. (2024) | 2.3k | * | 5 | ✓ | ✗ | ✗ |
| Ours | 5.48k | 20 | 13.5 | ✓ | ✓ | ✓ |

raw data in JSON format, a shared README file, the corresponding Python script, 17 general question-answer (QA) pairs: one description QA, one summary QA, five literal QAs, five inferential QAs, five reasoning QAs, 2 augmented QAs: 1 JSON-only QA, and 1 data-driven QA.

## 3.3 A NEW BENCHMARK FOR COMPREHENSIVE CHART UNDERSTANDING

A chart expert model should be capable of understanding a wide range of common chart types and, like a human, should not only be able to answer questions of varying complexity but also grasp the underlying data. However, as shown in Table 1, existing chart benchmarks either cover only a limited range of chart types (*e.g.*, line, bar, and pie charts) or lack comprehensive QA sets to evaluate a model's understanding of charts from various perspectives, including raw data comprehension, inferential abilities, and mathematical reasoning capabilities. To bridge this gap, we propose a comprehensive benchmark derived from the aforementioned synthetic dataset. It covers 20 different chart types, three different levels of QAs (literal, inferential, and reasoning QAs), and provides both long and short answers. Notably, the chart images in the benchmark are not all annotated, allowing assessment of the model's ability to understand the underlying data of a chart as humans do. To ensure the quality of the images in the benchmark, we employed human evaluations to filter the data and obtain a high-quality test set. The evaluations are based on two criteria: *Answerability*: whether the question is answerable given the chart image. *Correctness*: whether the provided answer is correct. Please refer to Sec. 3.3 in the supplementary materials for more details about benchmark statistics, filtering, analysis, etc. Note that these QAs equally cover literal, inferential, and reasoning questions for measuring chart understanding of MLLMs.

## 4 EXPERIMENTS AND MODEL ANALYSIS

### 4.1 EXPERIMENTAL SETUP

**Benchmark.** Our evaluation utilizes four classical benchmarks to compare our model against previous works. Specifically, we use the ChartQA dataset (Masry et al., 2022), which includes 1.5k chart images in its test set, divided into human-written and machine-generated questions with 1.2k QA pairs each. The human-written questions often require mathematical reasoning. ChartQA also provides CSV data for each image, enabling us to conduct a Chart-to-Table (or Chart Extraction) task to assess the ability of MLLMs to extract raw data from charts, following previous studies (Han et al., 2023; Liu et al., 2022a). Additionally, we use the PlotQA dataset Methani et al. (2020) where images generally lack numerical value annotations, necessitating value inference relative to the Y-axis. For evaluating the models' capability to capture global concepts, we assess on the Chart-to-Text task using the *Pew* and *Statista* splits from the dataset (Kantharaj et al., 2022b). The Pew split contains 9k images accompanied by descriptions written by professional editors, while the Statista split includes 33k images that often feature descriptive text within the charts themselves, making it easier than Pew.

**Metrics.** For ChartQA and PlotQA, we adopt the *relaxed accuracy* metric for numeric answers, allowing a 5% margin of error from the exact value, and use exact match for non-numeric answers as per the standard in previous studies (Masry et al., 2022; Han et al., 2023). In the Chart-to-Table task, we measure performance using *F1* score of *Relative Mapping Similarity* (RMS) and *Relative Number Set Similarity* (RNSS) to evaluate numeric accuracy and table similarity, respectively. For the Chart-to-Text task, we use *BLEU-4*, an N-gram matching score, following (Kantharaj et al., 2022b).

Table 2: **Ablation of stage-1 training.** This empirically verifies that pre-training basic chart visual perception is still important, even with abundant stage-2 instruction fine-tuning data. Moreover, learning to predict JSON data is beneficial, even on top of pre-training with descriptive captions.

| Training data | ChartQA | | Our benchmark | | |
|---|---|---|---|---|---|
| | human | augmented | literal | inferential | reasoning |
| LLaVA-CC3M-Pretrain pairs Liu et al. (2024b) | 44.80 | 83.92 | 41.45 | 34.09 | 22.31 |
| + Chart-description pairs | 48.56 | 86.89 | 42.71 | 33.68 | 23.51 |
| + Chart-JSON data pairs | **52.28** | **87.68** | **44.96** | **34.94** | **24.61** |

Table 3: **Ablation of stage-2 training.** Each type of new instruction / QA data improves the final performance consistently across almost all metrics. Best result is highlighted in **Bold** and the second best is underlined. [†] denotes inference technique without extra data.

| Training data | ChartQA | | Our benchmark | | |
|---|---|---|---|---|---|
| | human | augmented | literal | inferential | reasoning |
| LLaVA-Instruct-150K QAs | 45.84 | 86.48 | 16.54 | 15.99 | 6.57 |
| + description and summary QAs | 47.04 | **87.76** | 19.90 | 15.69 | 5.26 |
| + Literal / infer. / reasoning QAs | 48.96 | 87.52 | 40.55 | 33.33 | 21.30 |
| + JSON-only QAs | 49.60 | 87.36 | 41.45 | 34.84 | 22.36 |
| + Data-driven QAs | 52.28 | 87.68 | 44.96 | 34.94 | 24.61 |
| + Data Prompting[†] | **56.96** | 87.60 | **52.00** | **41.75** | **31.90** |

**The 3-stage Training Process.** Unlike previous approaches that convert a general MLLM into a chart-specific expert by only applying LoRA fine-tuning on limited high-quality data (Han et al., 2023), training CHOPINLLM unfolds in three stages, illustrated in Fig. 2 (b). The 3-stage training enables our model not only to understand chart QAs and downstream tasks but also to capture the underlying data, thereby achieving a fundamental understanding of charts. In the initial pre-training stage, we fix the ViT and LLM while training the projector from scratch using original LLaVA data alongside our newly generated chart-description and chart-json pairs. The second stage involves freezing ViT and jointly fine-tuning the projector and LLM with both original LLaVA QA pairs and our generated chart QA pairs, enabling the LLM to comprehend visual tokens and facilitate chart question answering. Finally, we apply LoRA fine-tuning to align the LLM's response distribution with the target downstream dataset. Each stage is carefully studied and the results are presented in the following subsections. In the following study, we ablate 1 stage at a time and use the full-training setting for the other 2 stages.

### 4.2 STAGE 1: PRE-TRAINING FOR CHART FEATURE ALIGNMENT

In the first training stage, the goal is to align visual and linguistic features so that visual data can be seamlessly translated into the textual domain for LLM comprehension. Employing a strategy from Liu et al. (2024b), we use a projector to translate visual features from ViT (Dosovitskiy et al., 2020) into the textual domain, training it with pairwise image-caption data to enhance its capability to capture visual information. We explore three configurations: utilizing only LLaVA CC3M Pretraining data,[2] combining LLaVA data with chart-description pairs, and using LLaVA data with both chart-description and chart-raw data pairs. The data for stage two training remains consistent across these settings, summary QAs, description QAs, three-level QAs, text-only QAs, and data-driven QAs, as depicted in Fig. 2 (b). In stage three, all models undergo LoRA fine-tuning on the downstream dataset, using LLaVA-7B as the baseline for this comparison. Results are detailed in Table 2.

**Dense data alignment is beneficial for both chart data comprehension and reasoning.** For chart images, chart-description pairs act as standard image-caption pairs. However, to more effectively bridge the visual-textual gap, we also utilize chart-json pairs that encompass the underlying numerical data and its schema of the charts. This approach not only aligns visual features with textual descriptions but also significantly enhances model performance, as demonstrated by improvements of approximately 2% in literal QAs and about 1% in reasoning skills, according to results in Table 2.

---

[2]https://huggingface.co/datasets/liuhaotian/LLaVA-CC3M-Pretrain-595K

Table 4: **Comprehensive evaluation across four chart benchmarks.** CHOPINLLM achieves best QA results on both (mostly) annotated benchmark, ChartQA, and non-annotated benchmark, PlotQA. H and A denote the human and augmented branch in ChartQA, respectively. Stat. represent the statista split. $^{\dagger}$: our reproduction using the official code. Note that for fair comparison, we don't use chain-of-reasoning in the inference. The best result is highlighted in **Bold** and the second underlined. # chart data denotes the number of pairwise chart data used in the training. A and S in the data source represent annotated data and synthetic data, respectively.

| Method | Data # | Data Source | ChartQA | | | Chart-to-Table | | Chart-to-Text | | PlotQA* | |
|---|---|---|---|---|---|---|---|---|---|---|---|
| | | | H | A | Avg. | F1 | RNSS | Pew | Stat. | v1 | v2 |
| Pix2struct Lee et al. (2023) | 80M | A | 30.50 | 81.60 | 56.00 | - | - | 10.30 | 38.00 | - | - |
| Matcha Liu et al. (2022b) | 16M | S+A | 38.20 | 90.20 | 64.20 | - | - | 12.20 | 39.40 | - | - |
| Unichart Masry et al. (2023) | 7M | S+A | 43.92 | 88.56 | 66.24 | 52.71 | - | 12.48 | 38.21 | - | - |
| DePlot Liu et al. (2022a) | 0.5M | S+A | - | - | - | 87.22 | 94.28 | - | - | - | - |
| LLaVA$_{7B}$$^{\dagger}$ Liu et al. (2024b) | - | - | 36.00 | 67.44 | 51.72 | 56.96 | 91.83 | 8.50 | 21.50 | 27.26 | 30.64 |
| LLaVA$_{13B}$ | - | - | 37.68 | 72.96 | 55.32 | 48.95 | - | 7.16 | 24.65 | - | - |
| LLaVA$_{13B}$$^{\dagger}$ | - | - | 42.56 | 73.60 | 58.08 | 63.18 | 93.18 | 8.83 | 22.39 | 27.68 | 30.98 |
| ChartLlama$_{13B}$ Han et al. (2023) | 0.16M | A | 48.96 | 90.36 | 69.66 | 89.84 | 94.65 | 14.23 | 40.71 | 29.76 | 29.93 |
| MMC$_{7B}$ Liu et al. (2024a) | 0.6M | S+A | - | - | 57.40 | - | - | - | - | - | - |
| ChartInstruct$_{7B}$ Masry et al. (2024a) | 0.19M | A | 45.52 | 87.76 | 66.64 | 18.87 | 34.59 | 13.83 | **43.53** | - | - |
| ChartAst$_{13B}$ Meng et al. (2024) | 24M | S+A | **65.9** | **93.9** | **79.9** | **91.6** | - | **15.5** | 41.0 | - | - |
| CHOPINLLM $_{7B}$ | 5M | S | 52.28 | 87.68 | 69.98 | 83.63 | 95.27 | 11.50 | 38.97 | 30.06 | 31.08 |
| CHOPINLLM $_{13B}$ | 5M | S | 54.11 | 88.67 | 71.39 | 88.12 | **95.95** | 12.66 | 40.81 | **33.98** | **33.96** |

## 4.3 STAGE 2: END-TO-END FINE-TUNING

The second stage, end-to-end fine-tuning, trains the MLLM to actually understand the aligned visual tokens so that it follows the user instruction and reason about the answer, on top of the inherent language capability from the original LLM. We utilize a significant number of image-QA pairs to jointly tune the LLM and the projector. To evaluate the effectiveness of incorporating chart QAs during fine-tuning, we conduct ablation studies starting with a baseline that uses only LLaVA Instruct-150K data,[3] incrementally adding extra QA pairs. All methods leverage the same pre-training weights, derived from training on LLaVA data with both chart-description and chart-raw data pairs (the best setting in Sec. 4.2). In stage three, all models undergo LoRA fine-tuning on the downstream dataset. Comprehensive results are presented in Table 3.

**JSON-only QAs allow transferring pure text reasoning abilities to multimodal chart understanding.** The chart understanding of MLLMs can be seen as two stages: visual and text raw data alignment (which is done in the training of the first stage) and question answering with reasoning ability on the raw textual data (JSON). Thus, with a well-aligned first stage training, we hypothesize that re-blending some pure textual QAs, preserving the ability of reasoning on text raw data, can also benefit the reasoning abilities in visual-text scenarios. As detailed in Sec. 3.2, for JSON-only QAs, rather than utilizing chart images and QAs, we replace the chart image with JSON data and a README, resulting in purely text-based QAs for training. Table 3 demonstrates the effectiveness of each QA type. We discover that re-blending JSON-only data during the end-to-end fine-tuning stage improves chart reasoning skills, matching the assumption.

**Data-driven QAs in the fine-tuning stage enable MLLMs to enhance prediction accuracy through data prompting.** As detailed in Sec. 3.2, data-driven QAs are multi-turn QAs, which require models to extract raw data before answering given questions. Combined with the raw data reasoning abilities enhanced via JSON-only QAs, the model can perform data prompting during inference, where models achieve better reasoning robustness by first extracting raw data and then answering the given question based on the data. Please refer to Appendix J for some examples. As shown in Table 3, data-driven QAs significantly enhance the model's ability to capture visual information. Furthermore, leveraging data prompting in inference significantly improves performance across all downstream tasks.

---

[3] https://huggingface.co/datasets/liuhaotian/LLaVA-Instruct-150K

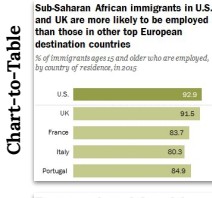

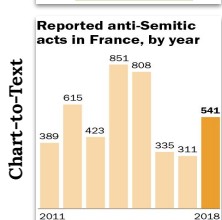

| | Gold table | | Ours | | ChartLlama | | LLaVA | |
|---|---|---|---|---|---|---|---|---|
| | Entity | Value | Entity | Value | Entity | Value | Entity | Value |
| | Portugal | 84.0 | Portugal | 84.9 | Portugal | 84.0 | Portugal | 84.9 |
| | Italy | 80.3 | Italy | 80.3 | Italy | 80.3 | Italy | 80.3 |
| | France | 83.0 | France | 83.7 | France | 83.0 | France | 88.7 |
| | UK | 915.0 | UK | 91.5 | UK | 915.0 | UK | 91.5 |
| | U.S | 92.9 | U.S | 92.9 | U.S | 92.9 | U.S | 92.9 |

**GT:** In France, home to Europe's largest Jewish population, reports of anti-Semitic incidents rose dramatically in 2018. There were 541 cases reported last year ... a 74% increase from 2017, according to .... And already in 2019, there have been several new high-profile anti-Semitic incidents, including ... in a Jewish cemetery.
**Ours:** France ... according to a 2018 report by the French government. The report, which tracked reported "acts" – including hate crimes, hate speech and anti-Semitic acts by individuals against people and property – found that 541 such acts occurred in 2018, up from 311 in 2017.
**ChartLlama:** The ... the number of reported anti-Semitic acts in France from 2011 to 2018. The data shows a significant increase in the number of reported acts from 2011 to 2018. In 2011, there were 432 reported acts, which increased to 783 in 2018. The number of reported acts peaked in 2015 with 851, and then decreased to 651 in 2016. The chart also indicates a slight increase in 2017 with 653 reported acts.

Figure 4: **Qualitative comparison on Chart-to-text and Chart-to-Table task.** CHOPINLLM effectively captures the underlying data and generates accurate chart descriptions with correct numerical values. Green denotes correct numbers, while red indicates mistakes. Best viewed in color. Note that the gold table is the noisy ground truth provided in ChartQA dataset.

## 4.4 STAGE 3: DOWNSTREAM FINE-TUNING

We build CHOPINLLM with the best setting based on the previous observation (the data used in each stage can be found in Fig. 2 (b)), and we compare CHOPINLLM with existing chart understanding approaches, including Pix2struct (Lee et al., 2023), Matcha (Liu et al., 2022b), Unichart (Masry et al., 2023), Deplot (Liu et al., 2022a), LLaVA (Liu et al., 2024b), and ChartLlama Han et al. (2023). The results are shown in Table 4.

**Classical question-answering on ChartQA.** We find that CHOPINLLM achieves the second best performance on ChartQA, as shown in Table 4. Notably, compared to the recent work of ChartAst, we use significantly less data, and most importantly, our training data is fully synthetic, requiring no additional human effort. In comparison to the third-best model, ChartLlama, we outperform it by $\approx 5\%$ on the human split of ChartQA. Note that the human split in ChartQA is more challenging than the augmented split, as it contains more reasoning questions, suggesting that CHOPINLLM is better at performing reasoning tasks.

**Raw data and global concept understanding.** As shown in Table 4, CHOPINLLM achieves the competitive F1 score and the highest RNSS result, indicating that CHOPINLLM can capture not only the structure but numerical values of raw data of chart images. We note that the performance on the chart-to-table task may have been saturated, as the images are mostly annotated. In this context, this primarily measures the OCR capability and does not assess the ability to capture the underlying data. As for the Chart-to-Text, shown in Table 4, CHOPINLLM performs comparable in the global concept capturing and can caption chart image with meaningful texts.

**Performance on unannotated chart images.** Most of the images in ChartQA (Masry et al., 2022) are annotated, which means the numerical values of data points are explicitly shown on the images. We observe that existing chart MLLMs, such as ChartLlama (Han et al., 2023), seem to heavily rely on this annotation for chart understanding, which is not ideal since real-world charts may be unannotated. We further evaluate them using the PlotQA dataset, and the results are shown in the last column of Table 4. Notably, since training previous models like ChartLlama on PlotQA is infeasible, we load the model weights as used in ChartQA and perform zero-shot prediction on PlotQA. The results show that our model performs significantly better ($\approx 3\%$ improvement) on unannotated chart images, suggesting that our methods with fundamental training rely less on numerical annotations. Note that the comparison with ChartAst and ChartInstruct is not included, as it was trained on PlotQA, which would affect the validity of the zero-shot predictions on PlotQA.

## 4.5 MORE MODEL ANALYSIS

**Qualitative Examples.** We provide a qualitative comparison of chart-to-text and chart-to-table tasks, with results depicted in Fig. 4. In the chart-to-table task, our model accurately captures values from chart images, unlike LLaVA and ChartLlama. It is important to note that the gold data tables for

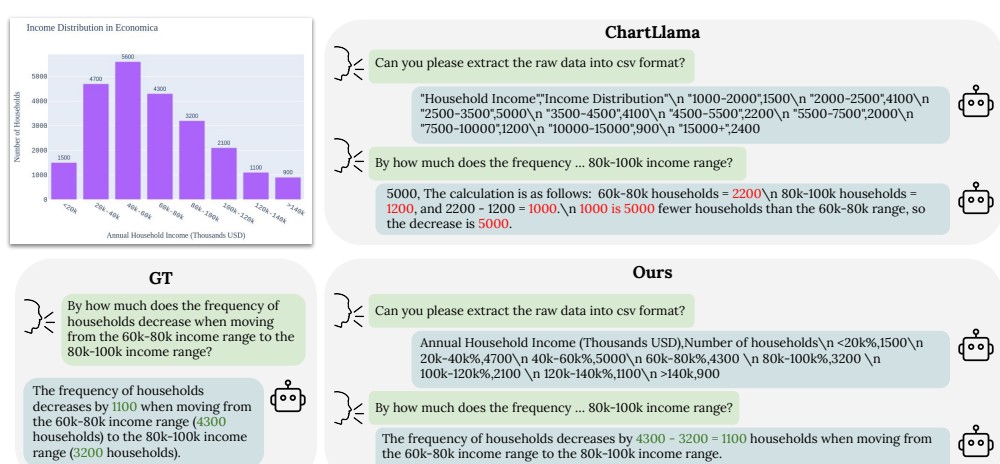

Figure 5: **Qualitative comparison of multi-turn chart question-answering.** Green denotes numbers that match groundtruth number, while red indicates mismatches. Best viewed in color.

Table 5: Performance comparison on different chart types. Overlapped denotes the chart types that are in both the ChartLlama training set and our dataset.

| Method | Basic | | | Overlapped | | | | | |
|---|---|---|---|---|---|---|---|---|---|
| | Line | Bar | Pie | Funnel | Gantt | Heatmap | Scatter | Box | Candle. |
| LLaVA Liu et al. (2024b) | 4.55 | 8.33 | 27.14 | 28.57 | 0.00 | 10.61 | 0.00 | 13.04 | 4.76 |
| ChartLlama Han et al. (2023) | 29.6 | 27.1 | 38.6 | 25.0 | 8.07 | 16.7 | 12.0 | **26.09** | 19.05 |
| CHOPINLLM | **40.9** | **29.2** | **68.6** | **60.7** | **15.8** | **25.8** | **26.0** | 21.7 | **23.8** |

ChartQA are not always directly accessible, leading to the use of existing models or OCR tools for data extraction. This process can introduce errors, such as misreporting the value 91.5 for the UK as 915.0, which can adversely affect the performance of MLLMs fine-tuned on such data. Despite these dataset inaccuracies, our model remains robust, correctly outputting values where ChartLlama does not. In the chart-to-text comparison, both ChartLlama and our model grasp the overall concept of the charts, but our model excels at accurately capturing and summarizing exact numerical values.

Additionally, as a multimodal chatbot, we emphasize preserving human-like multi-turn conversation abilities. Figure 5 presents a qualitative comparison on chart images with multi-turn QAs. Although ChartLlama extracts accurate numerical values, it fails to provide coherent explanations or reasonable text outputs. In contrast, CHOPINLLM not only extracts accurate data but also provides logical reasoning and coherent explanations, showcasing the effectiveness of our training approach.

**Performance Across Different Chart Types.** To asses ability of our model to perform chart understanding on a broader and more complex chart types we also evaluate it and state-of-the-art models on the proposed Benchmark discussed in Section 3.3. For an unbiased comparison, we focused on the short answer format in QA pairs to avoid variations in output preference. The results, detailed in Table 5, reveal that our model consistently outperforms the state-of-the-art across both overlapping and basic chart types. Notably, our benchmark, which features unannotated images, poses a greater challenge than ChartQA. The substantial performance improvement indicates that our model is adept at inferring data directly from charts and demonstrates superior reasoning capabilities.

## 5 CONCLUSION

In this paper, we explore the impact of fundamental training strategies in adapting generalist Multimodal Large Language Models (MLLMs) to chart understanding. We offer practical guidance for optimizing feature alignment pre-training and end-to-end fine-tuning. Leveraging these enhanced training strategies, we introduce a specialized chart MLLM, named CHOPINLLM, capable of interpreting diverse chart types independently of numerical annotations. Extensive experiments confirm that CHOPINLLM surpasses the previous state-of-the-art across four benchmarks, validating our framework's effectiveness. Additionally, we present a new benchmark specifically designed to evaluate MLLMs' comprehension across various chart types and multiple levels of understanding.

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
