# OpenReview forum: "On Pre-training of Multimodal Language Models Customized for Chart Understanding"
_ICLR.cc/2025/Conference — Submitted to ICLR 2025_

### Official Review · Reviewer_S9aK · 2024-10-26

**Soundness:** 3
**Presentation:** 3
**Contribution:** 3
**Rating:** 5
**Confidence:** 4

**Summary:**

This paper proposes a new Multimodal Large Language Model (MLLM), named CHOPINLLM, designed to enhance chart comprehension, especially in scientific and complex data visualizations. The model is tailored to bridge the gap between typical image-caption training data and chart-specific data, aiming to improve MLLM capabilities in extracting underlying numeric values from both annotated and unannotated charts. Furthermore, the paper also introduces a novel data generation pipeline to automaticaly produce large-scale pairwise data about chart understanding tasks. Finally, the paper construct a new benchmark comprising a diverse array of chart types and question-answering levels for robustly evaluate the chart understanding capabilities of MLLMs.

**Strengths:**

**Relevance**: The paper addresses a important task in MLLMs (Chart Understanding). The paper should be of interest that transcends the vision & language community to the broader research community.

**Novelty**:

- **Innovative Training Techniques**: The paper pioneers a set of training strategies (Three stages), notably using raw data in visual-language alignment, integrating text-only chart representations, and data-first reasoning in Q&A. These approaches contribute to making CHOPINLLM more adept at extracting and interpreting unannotated chart data, a significant advance over existing methods.
- **Data Generation Pipeline**: The paper propose a data generation pipeline, which addresses the challenge of obtaining diverse and high-quality chart data by using automated processes involving language models like GPT-4, which generate both the raw chart data and the Python code to produce chart images.

**Significance**:  This paper introduces a novel approach to training MLLMs, enabling accurate comprehension and reasoning over complex, unannotated charts, which significantly advances AI's ability to autonomously interpret data visualizations.

**Weaknesses:**

My primary concern about this paper is the performance of CHOPINLLM in chart understanding:

**Baselines**: This paper uses ChartAst [1] as its primary baseline. However, some baselines, such as TinyChart [2], ChartGemma[3], etc are being ignored. After going through and comparing these baselines on Chart QA, I don't find a significant performance advantage with CHOPINLLM.

**General MLLMs:** I don't get the practical significance of CHOPINLLM, the paper trained an MLLM by proposing a complex THREE STAGES TRAINING STRATEGY. For Stage 1, it's common to add CHART DATA to the image-text pair alignment stage, which has been used by several general MLLMs, e.g., LLama 3[4]. In Stages 2 and 3 (visual instruction tuning), adding chart QA data, cf. the above baselines is common. Therefore, I don't understand the significance of Contribution 1 shown in the Paper.

**Data Generation Pipeline:** By comparing with TinyChart, the automated pipeline proposed in the paper generates 5M of synthetic data, but the synthetic data generated by TinyChart is about 1M, and there is a big gap between the two in terms of performance on ChartQA (71.39 vs 82.88). This makes it hard to convince me that the data generation pipeline proposed in the paper is more efficient.

**Questions:**

Please see my feedback and suggestions above. I think the benchmark for CHART UNDERSTANDING presented in the paper is something that does promote the field, but for the first two contributions in the article, I don't see the obvious significance.

**Details Of Ethics Concerns:**

The paper does not require a separate ethics review

---

> ### Author Response · Authors · 2024-12-02
> **Rebuttal**
>
> > Some baselines, such as TinyChart [2], ChartGemma[3], etc are being ignored.
>
> We respectfully disagree with this comment, as it clearly violates the ICLR reviewing guidelines.
>
> ICLR rules state: "We consider papers contemporaneous if they are published within the last four months. That means, since our full paper deadline is October 1, if a paper was published (i.e., at a peer-reviewed venue) on or after July 1, 2024, authors are not required to compare their own work to that paper. Authors are encouraged to cite and discuss all relevant papers, but they may be excused for not knowing about papers not published in peer-reviewed conference proceedings or journals, which includes papers exclusively available on arXiv. Reviewers are encouraged to use their own good judgment and, if in doubt, discuss with their area chair."
>
> Asking or evaluating our paper with respect to papers in question violates these rules. Specifically, TinyChart was published at EMNLP in November 2024, which is AFTER the ICLR submission deadline of October 1st. OneChart and ChartGemma are preprints ONLY available on arXiv and have not been formally published to our knowledge. Thus we should not be faulted for not knowing and not comparing to these methods. Further, these papers have contributions that differ from ours, so while they may be achieving similar (or even better) results, they are doing it in fundamentally different manner, i.e., not invalidating our contributions.
>
> > What's the practical significance of CHOPINLLM? ... Adding CHART DATA to the image-text pair alignment stage has been used by several general MLLMs, e.g., LLama 3.
>
> Again, LLaMA 3 report was published in July 2024. Moreover, the details of how its raw data was formed are not thoroughly discussed or made available in that work. In contrast, our research begins with data generation and provides detailed information on the formats of the training QAs, serving as a valuable guide for future efforts in customizing MLLMs. Furthermore, in the second stage, key aspects such as JSON-only data and data-driven QA are not explored in the previous work on LLaMA. This is where our main contributions lie.
>
> > By comparing with TinyChart, there is a big gap between the TinyChart and ChopinLLM in terms of performance on ChartQA.
>
> As mentioned in an earlier response, comparison to TinyChart violates ICLR guidelines as the paper was published AFTER the ICLR submissions deadline.
>
> Further, TinyChart proposes the Program-of-Thought (PoT) paradigm, where VLM is trained to generate Python programs. Hence the synthetic data generated and the auto-generation pipeline is completely different. Our pipeline, data and observations are complementary. In the future, it may be possible to combine these paradigms and contributions to build a framework with even stronger capabilities. But again, this is outside the scope of our paper and opinion on our work should be made independent of TinyChart as outlined in ICLR rules.

---

### Official Review · Reviewer_3uLx · 2024-10-28

**Soundness:** 3
**Presentation:** 3
**Contribution:** 3
**Rating:** 5
**Confidence:** 5

**Summary:**

To enhance MLLM's ability to understand charts, the authors propose a process for generating charts and QA data and create a large training dataset. Based on this data, they introduce CHOPINLLM, a fine-tuned LLaVA-like model. Additionally, they propose a benchmark to evaluate the model's performance.

**Strengths:**

1. The authors present a clear and easy-to-understand workflow.
2. They provide a Chart instruction dataset that includes raw data and QA. The dataset creation process and its characteristics are well explained.
3. The authors offer a comprehensive summary and recommendations regarding MLLM training in the chart domain, particularly on instruction data selection and mixing.

**Weaknesses:**

1. Training aligned with raw data is already widely adopted (e.g., ChartAst, ChartReformer). Similarly, extracting chart data before QA has been explored (e.g., OneChart).
2. The authors emphasize that their model handles unannotated charts well, but there is no specific design for addressing it. Furthermore, results on unannotated charts are not provided. Benchmarked datasets like PlotQA are overly simple and repetitive, while others such as MMC, ChartBench, and ChartX (all are provided in Table 1) include higher-quality unannotated charts and QA, yet the authors do not report results on them.
3. Although the authors claim their method is MLLM-based fine-tuning, it’s unclear which base model they fine-tuned, making it difficult to evaluate the effectiveness of their data and training approach.
4. The experimental comparisons are insufficient. Some recent works like TinyChart, OneChart, and those mentioned in related works (e.g., ChartGemma) are not included in the comparative tables. Based on the numbers reported in those papers, CHOPINLLM’s results do not appear to be significant.

**Questions:**

1. The motivation for introducing the benchmark is unclear, as it appears similar in structure and evaluation to MMC without offering additional insights or conclusions.
2. The introduction needs smoother transitions; while the motivation and insights are understandable, it is difficult to follow how the problem is specifically addressed.
3. The authors should better clarify their contributions. While the workload is evident, the innovation is not, making the paper feel more like a technical report. For instance, if one contribution is the training method, does it generalize to other MLLMs like LLaVA, InternXC, Qwen, etc.?
4. After training heavily on chart-specific data, does the model's performance on other MLLM tasks (e.g., those in MME or SEED) degrade? How did the authors balance these aspects? Or is CHOPINLLM solely focused on chart-based QA?

---

> ### Author Response · Authors · 2024-12-02
> **Rebuttal (1/2)**
>
> Dear Reviewer 3uLx, We thank you for your review and valuable feedback. We appreciate your assessment of our work as "clear and easy to understand" and that "the dataset creation process and its characteristics are well explained." Please find detailed responses to each comment below.
>
> > Training aligned with raw data is already widely adopted, and extracting chart data before QA has been explored.
>
> We would like to note that previous works have focused on the raw data extraction task for basic chart types, while our work addresses a broader range of chart types and highlights the importance of raw data tasks in customizing generic multimodal language models. However, raw data extraction is not our primary contribution. Our key contributions are as follows: We introduce a Multimodal Large Language Model tailored for comprehensive chart understanding, excelling in interpreting various chart types, including unannotated ones. We propose a novel data generation pipeline that leverages text-only Large Language Models to efficiently produce large-scale pairwise data. We establish a robust benchmark that includes a diverse array of chart types and question-answering levels, specifically designed to rigorously evaluate MLLMs’ fundamental understanding of the scientific chart domain.
>
> > Performance comparison on other benchmark like MMC.
>
> As suggested, we further report the results of our model on the MMC benchmark. As shown in the table below, our model achieves the best performance on the Chart VQA task and the second-best performance overall (slightly lower than GPT-4V). These results further validate the effectiveness of our approach on unannotated charts.
>
> | Model       | MMC   |       |
> |-------------|-------|-------|
> |             | VQA   | MQA   |
> | LLaVA 1.5   | 0.24  | 0.51  |
> | MiniGPT-v2  | 0.21  | 0.47  |
> | mPLUG-owl   | 0.20  | 0.45  |
> | MMCA        | 0.26  | 0.56  |
> | GPT-4V      | 0.51  | **0.76**  |
> | Ours        | **0.54**  | 0.65  |
>
> > ...It’s unclear which base model they fine-tuned...
>
> In the Section B in the supplementary, we discuss the implementation details, where we mention the entire ChopinLLM is based on LLaVA model 7B and 13B version.
>
> > Some recent works like TinyChart, OneChart, and those mentioned in related works (e.g., ChartGemma) are not included in the comparative tables.
>
> We respectfully disagree with this comment, as it clearly violates the ICLR reviewing guidelines.
>
> ICLR rules state: "We consider papers contemporaneous if they are published within the last four months. That means, since our full paper deadline is October 1, if a paper was published (i.e., at a peer-reviewed venue) on or after July 1, 2024, authors are not required to compare their own work to that paper. Authors are encouraged to cite and discuss all relevant papers, but they may be excused for not knowing about papers not published in peer-reviewed conference proceedings or journals, which includes papers exclusively available on arXiv. Reviewers are encouraged to use their own good judgment and, if in doubt, discuss with their area chair."
>
> Asking or evaluating our paper with respect to papers in question violates these rules. Specifically, TinyChart was published at EMNLP in November 2024, which is AFTER the ICLR submission deadline of October 1st. OneChart and ChartGemma are preprints ONLY available on arXiv and have not been formally published to our knowledge. Thus we should not be faulted for not knowing and not comparing to these methods. Further, these papers have contributions that differ from ours, so while they may be achieving similar (or even better) results, they are doing it in fundamentally different manner, i.e., not invalidating our contributions.

---

> > ### Author Response · Authors · 2024-12-02
> > **Rebuttal (2/2)**
> >
> > > The motivation for introducing the benchmark is unclear.
> >
> > As shown in Table 1, our benchmark has the following unique features: (1) a wider variety of chart types with a fairly equal distribution across all types, (2) comprehensive evaluation for each chart image, and (3) raw JSON data provided for each image. The reviewer’s comment suggesting that our benchmark is similar in structure and evaluation to MMC is misleading, and here is our clarification: Most of the chart data in MMC consist of basic chart types (e.g., bar, line, and pie charts), whereas our benchmark has 20 different chart types. The structure of the two benchmarks is fundamentally different. In MMC, 73% of the QA structure comprises yes/no questions, while our benchmark includes open-form questions with answers that can be numerical values, yes/no, or other types. Most importantly, MMC does not provide raw data for each chart image in the benchmark, which limits its ability to evaluate a model's comprehension of raw data.
> >
> > > Clarification about the contribution of the training method.
> >
> > Our key contributions are as follows: We introduce a Multimodal Large Language Model tailored for comprehensive chart understanding, excelling in interpreting various chart types, including unannotated ones. We propose a novel data generation pipeline that leverages text-only Large Language Models to efficiently produce large-scale pairwise data. We establish a robust benchmark that includes a diverse array of chart types and question-answering levels, specifically designed to rigorously evaluate MLLMs’ fundamental understanding of the scientific chart domain. Regarding the training method, as detailed in Section B, CHOPINLLM is built upon the LLaVA model, incorporating a visual encoder, an adaptor, and an LLM, similar to Qwen.
> >
> > > Is CHOPINLLM solely focused on chart-based QA?
> >
> > In this paper, similar to previous works [1-3], we investigate strategies for customizing MLLMs for chart understanding tasks. The study of balancing customized and universal models is left as a future direction.
> >
> > [1] Meng, F., Shao, W., Lu, Q., Gao, P., Zhang, K., Qiao, Y. and Luo, P., 2024. Chartassisstant: A universal chart multimodal language model via chart-to-table pre-training and multitask instruction tuning. ACL main conference, 2024.
> >
> > [2] Liu, F., Wang, X., Yao, W., Chen, J., Song, K., Cho, S., Yacoob, Y. and Yu, D., 2023. Mmc: Advancing multimodal chart understanding with large-scale instruction tuning. NAACL, 2024.
> >
> > [3] Han, Y., Zhang, C., Chen, X., Yang, X., Wang, Z., Yu, G., Fu, B. and Zhang, H., 2023. Chartllama: A multimodal llm for chart understanding and generation. arXiv preprint arXiv:2311.16483.
> >
> > > Writing and typo in the introduction
> >
> > Thanks reviewer for pointing the issue out. We will revise it accordingly in the final version.

---

### Official Review · Reviewer_jKvL · 2024-10-29

**Soundness:** 2
**Presentation:** 4
**Contribution:** 2
**Rating:** 6
**Confidence:** 5

**Summary:**

The paper investigates the design space of chart understanding pretraining of multimodal LLMs along with a fully automatic synthetic data generation pipeline to resemble real-world charts. The resulting model, ChopinLLM, when pretrained on a mixture of LLaVA pretraining data and the synthetic data and fine-tuned on a mixture of LLaVA QAs and the synthetic QAs, achieves competitive performance on its own chart understanding benchmark and decent performance on a variety of other chart understanding benchmarks. The authors well documented the data generation pipeline and mappings between data usage at different stages and model performance.

**Strengths:**

- Investigating ways to improve chart understanding of MLLMs from the “pretraining” (e.g., aligning the connector with captioning data) perspective is rarely explored, which sets this work apart from others that focus on chart understanding in supervised finetuning of the full model on chart QAs. Experiments demonstrate that having a curated chart understanding dataset for pretraining can significantly enhance the model’s performance when later supervised finetuned on the same set of visual QA dataset.
- The paper is clearly written and examples of data and the data curation process are well documented in the supplementary materials.
- The experiments on the effectiveness of different types of chart understanding data are well investigated, where the major contribution factor toward the performance boost is to learn to translate the entire chart into textual data sources and learn to use the pattern for inference.

**Weaknesses:**

- A main argument from the paper seems to be that existing models could learn a shortcut that uses chart annotations to analyze the chart and answer questions (L73), while your methods result in a model that has less reliance (L478). Yet, there are no controlled experiments from the paper to support either claim.
- Lack of discussions and/or ablations on the effectiveness of orthogonal data and code generation compared to first generate the data then code. Generating code without knowing the data distribution/patterns limits the variations of the charts and may also create suboptimal layout of the charts. For example, if the data generator chooses to generate data that grows exponentially while the code generator chooses to create the corresponding axis in a linear scale, this can make the plot look awkward and it will also be hard to learn/interpret data from both human/model’s perspective. Some discussion and experiments on these scenarios (and how they could affect training) would be beneficial.
- Cost-effectiveness of data in terms of training is rarely discussed or compared with. While authors proposed a data pipeline that is cost-effective in synthesis, how much a fixed amount of data (or a fixed amount of compute) helps models learn chart understanding is not ablated. For example, when reducing ChartAst’s data to 5M, does model trained on your data perform better? Similarly, you can also reduce the amount of your training data to match the amount in ChartLlama, MMC or ChartInstruct and compare the performance.
- Adding chart-specific data to the pertaining dataset makes chart understanding data over-represented. As most multimodal LLMs tend to be used to solve a diverse range of tasks (i.e., not limited to chart understanding), it is unknown if such data imbalance affects models’ performance on other tasks that require visual perception and reasoning.
- I noticed that the most significant improvement of the performance on your benchmark happens when you add the same types of questions in stage-2 training, yet the performance gain on ChartQA is very small — which could indicate that your literal/inferential/reasoning QAs have a narrow and biased distribution. From a benchmarking perspective, this means that someone can easily gain huge performance boost by scaling up the amount of synthetic data under this distribution (which is easy to scale and can be fully automated as you documented), yet the models’ utility in real-world chart understanding can still remain low. I wonder if authors can provide some discussions on the validity of the numbers reported from your benchmark in terms of real-world chart understanding utility.

**Questions:**

- Line 300: The reference seems to be wrong (should be section C instead of 3.3?).
- Line 274: The “chart variation” terminology can be misleading without reading additional context e.g., it refers to having multiple styles of chart for the same data instead of the visual diversity of the charts.
- Line 478: I wonder if a formal ablation is conducted with respect to reliance on numerical annotations. A stronger performance on unannotated chart images does not necessarily indicate that the model doesn’t rely on numerical annotations. There are many possible reasons, such as questions on unannotated chart images tend to be easier, etc. A formal controlled study is warranted for the suggestion that your methods rely less on numerical annotations compared to a well-controlled baseline.
- LLaVA 1.5 only supports resolution up to 336^2. Have you considered the resolution bottleneck for your training experiments and evaluations. Does training on your data become more effective if you scale up the training resolution?
- Line 522: there is one typo.
- Line 342: I interpret Data-driven QAs as the finetuning data for model to generate the JSON before giving the answer, and Data Prompting is a natural language prompt applied during inference time to elicit generation of the JSON before giving the answer. Is my interpretation correct? Does the model generate the JSON when there is no explicit prompting but when there are data-driven QAs?

---

> ### Author Response · Authors · 2024-12-02
> **Rebuttal (1/3)**
>
> Dear Reviewer jKvL, we would like to thank you for your review and valuable feedback. We are pleased to hear that you found the article "clearly written," that "examples of data and the data curation process are well documented," and that "the experiments on the effectiveness of different types of chart understanding data are well investigated." Please find detailed responses to each comment below.
>
> > There are no controlled experiments from the paper to support the claim that the proposed methods has less reliance on the chart annotations. ...
>
> We thank the reviewer for highlighting this concern and are pleased to provide additional results to support our claim. First, we would like to clarify that in Table 4, we compare results on both non-annotated charts (PlotQA) and annotated charts (ChartQA). Our model consistently outperforms previous works, demonstrating its effectiveness in reducing reliance on annotated data compared to prior methods. Additionally, we utilized data from our benchmark to further validate this with more controlled experiments as suggested. Specifically, in our dataset, chart images are generated using Python scripts. To set up controlled experiments, we selected a Python script from the bar chart split and modified it to create both annotated and non-annotated versions. We then generated bar chart images with and without annotations by applying the scripts to the corresponding generated JSON raw data, resulting in 382 chart image-QA pairs for the experiments. We ran our model and compared its performance with previous works. The results are shown in the table below. From the results, we observe that previous models, such as Phi 3.5, ChartLLama, and ChartAssistant, experience a performance drop when annotations (i.e., the exact values for each bar) are removed from the chart images. In contrast, our method performs even better without annotations, verifying our claim that our approach is less reliant on annotated data.
>
> | Model       | w/ anno          |            |         | w/o anno |               |      |           |
> |-------------|------------------------------|---------------------|-------------------|-------------------|---------------------------------|---------------------|-----------|
> |             | Literal                      | Inferential         | Reasoning                      | Literal             | Inferential | Reasoning |
> | InternVL2   | 55.26                        | 63.64               | 31.03                          | 89.47               | 87.10       | 32.26     |
> | Phi-3.5-V   | 73.53                        | 80.00               | 22.58                          | 62.86               | 71.43       | 26.67     |
> | ChartLlama  | 12.90                        | 33.32               | 6.67                           | 0.00                | 33.33        | 0.00      |
> | ChartAst    | 44.12                        | 48.65               | 10.34                          | 40.00               | 43.33       | 16.67     |
> | Ours        | 43.02                        | 43.75               | 25.00                          | 48.57               | 65.38       | 39.00     |
>
> > Lack of discussions and/or ablations on the effectiveness of orthogonal data and code generation compared to first generating the data then code. Generating code without knowing the data distribution/patterns limits the variations of the charts and may also create suboptimal layout of the charts. ...
>
> We thank the reviewer for pointing out this issue and are glad to provide further discussion on our data generation process. First, we would like to clarify an implementation detail: the code generation process is not entirely blind to the raw data. Specifically, we first generate a small batch of raw data (e.g., 10 samples) and include these samples in the prompt when generating the Python scripts. We apologize for omitting this detail in the original submission and will ensure it is included in the final version. By incorporating this small batch of samples during code generation, we did not observe a significant occurrence of suboptimal layouts in the generated charts. Furthermore, we would like to point out that sequential generation can introduce more bias, as each subsequent step in the sequence might reinforce previous errors or biases in the generated data. In contrast, an orthogonal approach, which generates data independently or with diversified prompts, can mitigate this issue by ensuring a broader and more varied exploration of the data space.

---

> > ### Author Response · Authors · 2024-12-02
> > **Rebuttal (2/3)**
> >
> > > Discussion about the cost-effectiveness of data in terms of training.
> >
> > We thank the reviewers for raising this practical concern. We address the cost-effectiveness of data in terms of training in Section D2. In this section, we examine the cost-effectiveness of the data following a common scaling law experiment protocol [1]. Specifically, we analyze the log-linear relationship between FLOPs and parameters, as well as FLOPs and training tokens, to determine the optimal number of training data points for a specific model. This strategy, widely adopted in existing works such as LLaMA 3, is particularly useful because training large-scale models multiple times is computationally expensive. By employing this approach, researchers can infer the optimal amount of training data by experimenting with smaller-scale models. Based on the findings from these studies, we determined the optimal number of training data points to be 5 million. However, it is important to note that in these scaling experiments, we focus exclusively on training with synthetic data, as the primary objective of this paper is to explore the feasibility of using synthetic data to customize models. The study of using a broader range of data, including real, human-labeled, and synthetic data, falls outside the scope of this work.
> >
> > [1] Kaplan, J., McCandlish, S., Henighan, T., Brown, T.B., Chess, B., Child, R., Gray, S., Radford, A., Wu, J. and Amodei, D., 2020. Scaling laws for neural language models. arXiv preprint arXiv:2001.08361.
> >
> > > Adding chart-specific data to the pertaining dataset makes chart understanding data over-represented.
> >
> > We thanks reviewer for pointing out this potential issue. In this paper, similar to previous works [1-3], we investigate strategies for customizing MLLMs for chart understanding tasks. The study of balancing customized and universal models is left as a future direction.
> >
> > [1] Meng, F., Shao, W., Lu, Q., Gao, P., Zhang, K., Qiao, Y. and Luo, P., 2024. Chartassisstant: A universal chart multimodal language model via chart-to-table pre-training and multitask instruction tuning. ACL main conference, 2024.
> >
> > [2] Liu, F., Wang, X., Yao, W., Chen, J., Song, K., Cho, S., Yacoob, Y. and Yu, D., 2023. Mmc: Advancing multimodal chart understanding with large-scale instruction tuning. NAACL, 2024.
> >
> > [3] Han, Y., Zhang, C., Chen, X., Yang, X., Wang, Z., Yu, G., Fu, B. and Zhang, H., 2023. Chartllama: A multimodal llm for chart understanding and generation. arXiv preprint arXiv:2311.16483.
> >
> > > Why the most significant performance improvement on the proposed benchmark happens after adding the literal/inferential/reasoning questions in stage-2 training?
> >
> > We thank the reviewer for pointing out this concern, and we are glad to provide further details. We note that even in the third stage of training, where we perform LoRA fine-tuning on ChartQA, the model’s QA capabilities on general chart types remain limited. This is because ChartQA primarily includes basic chart types, whereas our benchmark contains data from a broader range of general chart types. Consequently, before introducing literal, inferential, and reasoning QAs, the model has limited knowledge to answer questions about chart types beyond the basic ones. Regarding the experiment on biased distribution analysis and real-world utility, we currently do not have a clear idea of how to test this, as our benchmark has already undergone human filtering. We would greatly appreciate any suggestions or recommendations from the reviewer on potential experiments to address this concern.

---

> > > ### Author Response · Authors · 2024-12-02
> > > **Rebuttal (3/3)**
> > >
> > > > A formal controlled study is warranted for the suggestion that your methods rely less on numerical annotations compared to a well-controlled baseline.
> > >
> > > Please refer to the response for the first question.
> > >
> > > > Have you considered the resolution bottleneck for your training experiments and evaluations?
> > >
> > > Due to computational constraints, we were unable to study the effect of using larger image resolutions. However, higher resolution typically leads to improved performance.
> > >
> > > > Interpretation of data-driven QAs. Does the model generate the JSON when there is no explicit prompting but when there are data-driven QAs?
> > >
> > > Yes, your interpretation is correct. For data-driven QAs, it involves a multi-turn conversation where there is an explicit prompt to extract raw data before addressing the chart-related question.
> > >
> > > > Writing and typo
> > >
> > > Thanks for pointing out the writing issue. We will revise them accordingly in the final version.

---

> > ### Comment · Reviewer_jKvL · 2024-12-02
> >
> > Thank you for your response! For the annotation experiments, I am curious why 2/5 models i.e., InternVL2 and yours show a significant performance degradation when annotations are provided, assuming the annotated values are correct? I am curious about your thoughts on this. It doesn't look like a robustness issue since InternVL2 appears to not get trained on your data only and it serves as a general purpose model.

---

> > > ### Author Response · Authors · 2024-12-02
> > > **Response to Reviewer jKvL**
> > >
> > > Thank you for responding to the rebuttal and for expressing your curiosity about the additional results.
> > >
> > > We also observed this interesting phenomenon and hypothesize that it could be due to biases in the training data. Specifically, Phi-3.5-V and ChartAst may possess better OCR capabilities, potentially as a result of more extensive OCR training data, compared to InternVL and our model. This difference might lead the models to interpret chart images in distinct ways.
> > >
> > > Upon analyzing the results of InternVL and our model versus Phi-3.5-V and ChartAst, we found that chart annotations can potentially introduce noise for InternVL. This often causes the model to generate incorrect answers by misidentifying the target portion of the chart. In contrast, we observed that the Phi-3.5-V model excels in OCR, enabling more accurate predictions on annotated charts.
> > >
> > > Such biases or preferences can result in significant performance variations particularly on the first two levels of questions as these require the ability to capture both local and global information from the given charts.

---

> > > > ### Comment · Reviewer_jKvL · 2024-12-02
> > > >
> > > > Thank you for sharing your thoughts! I believe this could be an interesting finding worth investigating more. I hope the authors can continue analyzing these patterns in their paper to justify the claim. Regardless, given authors have shown the general case with results (where models appear to perform better with annotations, which could serve as a shortcut), I have raised my score to 6.

---

### Official Review · Reviewer_8qKC · 2024-11-01

**Soundness:** 2
**Presentation:** 3
**Contribution:** 2
**Rating:** 5
**Confidence:** 4

**Summary:**

The paper introduces a pipeline to create a comprehensive dataset for fine-tuning the proposed MLLM, CHOPINLLM for chart understanding. It highlights that incorporating raw data values during pre-training, substituting images with textual data in fine-tuning, and prioritizing data extraction before answering questions significantly improve performance. Additionally, a benchmark dataset is developed to evaluate MLLMs’ comprehension of various chart types across different complexity levels.

**Strengths:**

1. The paper introduces efficient training techniques that significantly enhance chart comprehension.
2. CHOPINLLM, a model for chart understanding, demonstrates strong performance with various chart types.
3. A benchmark is established to evaluate MLLMs' comprehension of different chart types, aiding future research.
4. The data generation pipeline uses text-only Large Language Models to efficiently create diverse datasets, reducing costs and complexity.

**Weaknesses:**

1. CHOPINLLM did not achieve state-of-the-art (SOTA) performance in Table 4. While the authors claim that higher-performing models benefited from using more data and annotated datasets, there is no evidence showing that the proposed synthetic data offers performance gains when combined with existing datasets. Demonstrating that such a combination improves results would strengthen the contribution of the synthetic data. Otherwise, the benefit of using only synthetic data to build an underperforming model appears limited. (this is my major concern)
2. The paper lacks comparisons with a broader range of SOTA MLLMs that are not specifically tailored for chart understanding, such as InternVL2 and Phi-3.5-V.
3. It omits comparisons with proprietary SOTA models like GPT-4o and Claude-3.5-Sonnet, which would help illustrate performance differences between open-source and proprietary models.

**Questions:**

In addition to the weaknesses:

1. What is the difference between annotated data and synthetic data that could be the major cause of the performance gap between CHOPINLLM and ChartAst-13B? What challenges exist in create synthetic data in comparable quality?
2. Can the data generation method be generalized to other domains where annotated data is harder to obtain? Demonstrating this would help justify the advantage of using only synthetic data for training and emphasize its broader applicability.

---

> ### Author Response · Authors · 2024-12-02
> **Rebuttal (1/2)**
>
> Dear Reviewer 8qKC, we would like to thank you for your kind review and valuable feedback. We sincerely appreciate your recognition of our work, noting its "strong performance with various chart types," the value of "our benchmark in aiding future research," and the potential of "our data generation pipeline to reduce costs and complexity." Please find our detailed responses to your comments below.
>
> > ...there is no evidence showing that the proposed synthetic data offers performance gains when combined with existing datasets. ...
>
> We thank the reviewer for raising this important concern. We would like to clarify that Table 4 demonstrates the performance gains achieved when combining the proposed synthetic data with existing datasets. Specifically, the LLaVA model in Table 4 was LoRA fine-tuned solely with ChartQA, whereas CHOPINLLM was also fine-tuned using ChartQA combined with our synthetic dataset (synthetic data for first two stages and ChartQA for last stage). By comparing the performance of LLaVA and CHOPINLLM, it is evident that our approach offers an improvement due to the inclusion of the synthetic data. We also emphasize the focus of this work and the strengths of using synthetic data. Our primary goal is to investigate whether synthetic data can be effectively utilized for training, offering two significant benefits: (1) enabling support for a broader range of chart types and (2) facilitating alignment training using raw data. Currently, there is no existing dataset that includes a large number of chart images encompassing various chart types, which highlights the importance and uniqueness of our synthetic dataset.
>
> > The paper lacks comparisons with a broader range of SOTA MLLMs, including models like InternVL2 and Phi-3.5-V, as well as proprietary models like GPT-4o and Claude-3.5-Sonnet, which could highlight performance differences between open-source and proprietary approaches.
>
> We report the results of GPT-4o, InternVL, and Phi-3.5V tested on ChartQA and our benchmark. Our model achieves comparable performance to GPT-4o and the state-of-the-art (SOTA) generic MLLMs. However, we note that a direct comparison with these SOTA MLLMs is not entirely fair, as the training data and computational costs differ significantly. For instance, Phi-3.5V requires 256 A100-80G GPUs for 6 days of training on 500 billion tokens (including vision and text tokens), whereas our model requires only 2 days of training with 8 A100 GPUs and significantly fewer training tokens.
>
> | Model       | ChartQA | Our benchmark (Literal) | Our benchmark (Inferential) | Our benchmark (Reasoning) |
> |-------------|---------|---------|-------------|-----------|
> | GPT-4o      | 64.0*   | 47.6    | 59.4        | 26.2      |
> | InternVL2   | 72.6    | 46.3    | 65.4        | 20.9      |
> | Phi-3.5-V   | 81.8    | 46.0    | 65.7        | 20.3      |
> | Ours        | 71.4    | 44.8    | 58.2        | 21.2      |
>
> *Number is obtained from Phi-3.5v paper

---

> > ### Author Response · Authors · 2024-12-02
> > **Rebuttal (2/2)**
> >
> > > What is the difference between annotated data and synthetic data that could be the major cause of the performance gap between CHOPINLLM and ChartAst-13B?
> >
> > There are two key differences in the data that could influence performance: (1) Amount of data: The majority of annotated and synthetic data (20M) collected for ChartAst-13B pertains to basic chart types, such as line, bar, and pie charts, which align with the benchmarks used in Table 4.  However, this data does not generalize well to other chart types, as demonstrated in Figure 8 in the supplementary material. (2) Quality of data: The data in ChartAst is human annotated which results in higher quality. Overall, we outperform ChartAst-13B by 15% in performance on a more comprehensive chart benchmark.
> >
> > > What challenges exist in creating synthetic data in comparable quality?
> >
> > Regarding the challenges associated with creating synthetic data, as described in Section A.1, the entire dataset is generated using GPT-4. However, large multimodal language models (MLLMs) like GPT-4 can occasionally make errors in generating raw data and Python scripts. To address this, we employ several filtering techniques, including format filtering, Python error filtering, and OCR filtering, to enhance the quality of the generated data. Please refer to Section A.1 for more details. To further evaluate the quality of the generated data, we conducted a human study to analyze the accuracy of our generation pipeline. Human evaluators assessed the validity of the generated image-QA pairs based on two criteria: (1) whether the answer could be derived from the given image and (2) whether the generated answer was correct. A generated image-QA pair was considered valid only if both criteria were met. We collected responses for 150 image-QA pairs, and the validity percentages for each QA level were as follows: literal – 82%, inferential – 88%, reasoning – 92%. Note that this analysis pertains to the training data.
> >
> > > Can the data generation method be generalized to other domains where annotated data is harder to obtain?
> >
> > In this paper, we focus solely on chart data, as it is one of the most widely used representations for data visualization. However, our proposed data generation pipeline can be applied to any type of data that can be generated using Python code. For example, it can handle charts, tables, tables with charts, geospatial maps, and documents. These structured data types can be generated using Python libraries such as Matplotlib, Plotly, and others. Specifically, for geospatial data, one can synthesize raw data with random geographic distributions, and the generated Python code can then visualize these data in 2D or 3D representations.

---

### Meta-Review · Area_Chair_c8gb · 2024-12-22

**Metareview:**

The paper introduces CHOPINLLM, a multimodal large language model designed to enhance chart comprehension, particularly for unannotated charts. It presents a data generation pipeline that automatically produces a synthetic dataset tailored for chart understanding tasks and proposes a new benchmark to evaluate performance across diverse chart types and question-answering levels.

However, Reviewers noted a lack of demonstrated performance advantages over existing state-of-the-art models. The paper does not provide sufficient evidence that the proposed synthetic data improves performance when combined with existing datasets. There are also concerns about the model's generalizability beyond chart-specific tasks and its potential over-reliance on chart-specific data. Given these substantial issues related to originality, effectiveness, and broader impact, the paper does not meet the acceptance criteria at this time.

**Additional Comments On Reviewer Discussion:**

During the rebuttal period, Reviewer 8qKC reiterated concerns about insufficient evidence that the synthetic data improves performance with existing datasets and the lack of broader state-of-the-art comparisons. The authors presented additional results and explained omissions due to conference guidelines, but Reviewer 8qKC maintained a negative stance.

Similarly, Reviewers 3uLx and S9aK raised questions about the novelty and practical significance of the work, noting similarities to existing approaches and expressing doubts about the model's generalizability beyond chart-specific tasks. The authors briefly defended their approach, emphasizing the uniqueness of their data generation pipeline and compliance with submission policies, but the reviewers remained unconvinced and upheld their negative evaluations.

Weighing these points, it was determined that the paper does not sufficiently meet the standards for acceptance at this time.

---

### Decision · Program_Chairs · 2025-01-22

Reject